# Genomic Analysis and Functional Validation of Bidirectional Promoters in Medaka (*Oryzias latipes*)

**DOI:** 10.3390/ijms252413726

**Published:** 2024-12-23

**Authors:** Jingjie Liang, Yan Huang, Jiangling Li, Ruoxue Chen, Yanlong Lin, Haiqing Li, Xiangrui Cao, Tiansheng Chen

**Affiliations:** State Key Laboratory of Mariculture Breeding, Engineering Research Center of the Modern Technology for Eel Industry, Ministry of Education, Key Laboratory of Healthy Mariculture for the East China Sea, Ministry of Agriculture and Rural Affairs, Fisheries College of Jimei University, Xiamen 361021, China; 202261000039@jmu.edu.cn (J.L.); 202161000121@jmu.edu.cn (Y.H.); 202211908043@jmu.edu.cn (J.L.); 202211710036@jmu.edu.cn (R.C.); linyanlong1203@163.com (Y.L.); 202221062068@jmu.edu.cn (H.L.); 202121061025@jmu.edu.cn (X.C.)

**Keywords:** bidirectional promoters, gene regulation, medaka, multigene expression

## Abstract

Bidirectional promoters (BDPs) regulate the transcription of two adjacent, oppositely oriented genes, offering a compact structure with significant potential for multigene expression systems. Although BDPs are evolutionarily conserved, their regulatory roles and sequence characteristics vary across species, with limited studies in fish. Here, we systematically analyzed the distribution, sequence features, and expression patterns of BDPs in the medaka (*Oryzias latipes*) genome. A total of 1737 divergent gene pairs, representing 13% of medaka genes, were identified as potentially regulated by BDPs. These genes are enriched in essential biological processes, including organelle function, RNA processing, and ribosome biogenesis. Transcriptomic analysis revealed that co-regulation (co-expression and co-silencing) is a prominent feature of these gene pairs, with variability influenced by tissue and sex. Sequence analysis showed that medaka BDPs are compact, with most fragments under 400 bp and an average GC content of 42.06%. Validation experiments confirmed the bidirectional transcriptional activity of three histone-related BDPs in both medaka SG3 cells and embryos, demonstrating effective and robust regulatory efficiency. This study enhances our understanding of the genomic organization and transcriptional regulation in fish and provides a valuable reference for developing species-specific multigene expression systems in fish genetic engineering.

## 1. Introduction

Promoters are essential regulatory elements that modulate gene expression by influencing transcription efficiency. Divergent gene pairs, also referred to as head-to-head gene pairs, represent a unique genomic arrangement in which two adjacent genes, located on opposite DNA strands, are transcribed in opposite directions from a shared promoter region, termed a bidirectional promoter (BDP). These promoters are characterized by transcription start sites (TSS) of the two genes being separated by fewer than 1000 base pairs [1]. BDPs are widespread across species, from prokaryotes to eukaryotes, and are frequently associated with genes involved in conserved biological processes, such as the cell cycle, DNA repair, and chromatin remodeling [2,3,4,5,6]. By enabling the coordinated transcription of adjacent genes, BDPs provide an efficient mechanism for regulating functionally related processes, ensuring genomic stability, and facilitating cellular adaptability to environmental or physiological changes.

The functional capabilities of BDPs are closely linked to their unique sequence characteristics. Enriched in GC content and CpG islands, BDPs maintain an open chromatin state, enhancing RNA polymerase II binding efficiency [7,8]. Unlike unidirectional promoters, BDPs typically lack TATA boxes and instead rely on alternative core promoter elements, such as INR, DPE, and BRE, to initiate transcription [9]. Specific transcription factor binding sites (TFBS), including GABPA, SP1, and E2F, are more frequently associated with BDPs, suggesting their role in facilitating the coordinated transcription of adjacent genes [10]. Additionally, motifs responsive to environmental or tissue-specific stimuli, such as low-temperature responsive elements in *Arabidopsis thaliana* [11] and estrogen response elements in MCF-7 breast cancer cells [12], further underscore their role in dynamic responses to external signals. Furthermore, the CpG islands within BDPs are often hypomethylated, ensuring sustained transcription and preventing gene silencing, particularly for housekeeping or stress-responsive genes [10].

The ability of BDPs to regulate two adjacent genes simultaneously has fueled growing interest in their application for multigene co-expression systems. With their streamlined sequence architecture and precise regulatory control, natural BDPs have been successfully utilized in model organisms such as *Arabidopsis thaliana* and yeast, demonstrating their potential in synthetic biology and genetic engineering [11,13,14]. Despite these advancements, much about BDP-mediated regulation remains poorly understood. While the arrangement of divergent gene pairs is evolutionarily conserved across species, the sequence structure and regulatory functions of BDPs vary significantly, particularly between vertebrates and invertebrates [15]. These differences reflect distinct evolutionary pressures and highlight the diversity of gene regulatory mechanisms. Further investigation into the regulatory mechanisms of BDPs will enhance our understanding of genome regulation complexity and provide a theoretical foundation for advancing multigene expression technologies. As a transitional group between invertebrates and vertebrates, fish serve as a valuable model for studying the evolutionary conservation and divergence of BDPs. However, current research on fish BDPs remains sparse, emphasizing the urgent need for more comprehensive and systematic studies in this area.

In this study, we utilized the model organism medaka (*Oryzias latipes*) to systematically analyze the distribution of divergent gene pairs within its genome and investigate the sequence characteristics of BDPs. Additionally, we experimentally validated the activity of three histone-related BDP (H-BDP) fragments. This research provides novel insights into the distribution and regulatory mechanisms of BDPs in fish while also laying the foundation for understanding their evolutionary significance and advancing the development of exogenous gene expression tools for fish genetic engineering.

## 2. Results

### 2.1. Screening of Divergent Gene Pairs

In the medaka genome (ASM223467v1), we analyzed 26,870 genes and identified 1737 divergent gene pairs, encompassing 3499 genes and representing 13% of the genome (Appendix A). These pairs were categorized based on their 5′ end configuration: 252 pairs (20.3%) exhibit overlapping 5′ ends, while the remaining 1385 pairs (79.7%) do not (Figure 1A, B). Chromosome-specific distributions are detailed in Table 1. To assess the correlation between divergent gene pair distribution and gene density, a Pearson correlation coefficient of 0.13 was determined, indicating a lack of significant association (Figure 1C).

### 2.2. Functional Enrichment Results

Functional enrichment analysis of divergent gene pairs in the medaka genome revealed their significant roles in various biological processes. In the GO enrichment analysis (Figure 2A), highly enriched Biological Process (BP) terms included organelle assembly, ncRNA processing, RNA splicing, and ribonucleoprotein complex biogenesis. Notably, genes involved in cellular component organization and RNA processing were predominant. For Cellular Component (CC), mitochondrial-related terms, such as mitochondrial inner membrane, mitochondrial membrane, and mitochondrial envelope, were significantly enriched. In the Molecular Function (MF) category, nucleotide binding and nucleoside phosphate binding were the most abundant terms. In the KEGG pathway enrichment analysis (Figure 2B), pathways such as ribosome, oxidative phosphorylation, and nucleocytoplasmic transport were significantly enriched, highlighting the crucial roles of these divergent promoter genes in cellular component organization, metabolic processes, and ribosome biogenesis.

### 2.3. Expression Patterns of Divergent Gene Pairs in Medaka

To investigate the expression patterns of divergent gene pairs across different tissues, we analyzed transcriptomic data from both our laboratory and public databases, covering the ovary, testis, brain (GSE159188), and liver (GSE252744). Expression states were categorized into co-expressed, unidirectionally expressed, and co-silent (see Section 4, Figure 3, and Appendix A).

In the ovary and testis, co-expressed gene pairs were the most abundant, accounting for 50.2% and 50.8% of the total, respectively, while unidirectionally expressed pairs constituted 45.9% in the ovary and 45.1% in the testis. Co-silent pairs made up a minor fraction (3.9% in the ovary, 4.1% in the testis), indicating a strong prevalence of shared transcriptional regulation in gonadal tissues. Similarly, in the brain, co-expressed pairs were dominant, comprising 52.2% in females and 51% in males, while unidirectionally expressed pairs accounted for 45.6% and 46%, respectively. Co-silent pairs remained consistently low (2.2% and 3%), reflecting active transcriptional regulation and low levels of gene silencing.

In contrast, the liver showed a slightly different trend. Co-expressed gene pairs were less prevalent compared to other tissues, accounting for 40.2% in females and 36.8% in males. Unidirectionally expressed pairs dominated in the liver, making up 58.2% in females and 61.2% in males, while co-silent pairs were rare (1.6% and 2%, respectively). This lower proportion of co-expressed pairs suggests that divergent gene pairs in the liver may be regulated through alternative mechanisms or exhibit lower reliance on shared bidirectional promoters.

The overlap analysis revealed distinct patterns for co-expressed and co-silent gene pairs across male and female tissues (Figure 4, Appendix A). In male tissues, 170 gene pairs were co-expressed across the brain, testis, and liver, with additional tissue-specific pairs (123 in the brain, 123 in the testis, and 68 in the liver). Similarly, 217 gene pairs were shared across female tissues, with subsets specific to the brain (105), ovary (75), and liver (82). This suggests that co-expression is a more conserved regulatory mechanism across tissues. Co-silent pairs showed minimal overlap, with only 1 pair shared across male tissues and 4 pairs across female tissues. Most co-silent pairs were tissue-specific, such as 24 pairs in the testis and ovary and 6 pairs in the liver for both sexes.

### 2.4. Sequence Characteristics of Medaka BDPs

Next, we analyzed the sequence characteristics of 1,737 BDPs in medaka. The length distribution of BDP sequences is shown in Figure 5A, with 79.74% of the fragments originating from non-overlapping divergent gene pairs. Notably, 49.68% of these fragments were shorter than 400 bp. The GC content analysis revealed an average GC content of 42.06% and a median of 39.82% for BDPs. In comparison, non-BDP fragments had an average GC content of 38.43% and a median of 37.76%. As shown in Figure 5B, the proportion of BDP fragments with moderate GC content (40–60%) was significantly higher than that of non-BDPs, which were predominantly found in the lower GC content range (20–40%).

### 2.5. Validation of BDP Activity

To identify effective and stable BDP fragments in medaka, we performed a targeted screening based on transcriptome data from medaka tissues. This analysis was aimed at selecting gene pairs with strong, balanced expression profiles suitable for functional validation. Through this approach, histone-related gene pairs emerged as promising candidates due to their high expression levels and balanced expression ratios, indicating potential co-expression.

To assess the transcriptional initiation activity of these BDP fragments, we selected three histone-related gene pairs (Table 2). Due to the proximity of their TSS, the sequence between the ATG of each pair was amplified as histone-related BDPs (H-BDPs) to ensure promoter region coverage. These fragments were cloned into plasmids with GFP and RFP reporters (Figure 6A) and tested for BDP activity via transient transfection in SG3 cells. As shown in Figure 6B, all three fragments (H-BDP1, H-BDP2, H-BDP3) successfully drove GFP and RFP expression, confirming bidirectional activity in vitro. Quantitative analysis (Figure 6C) revealed H-BDP1 exhibited the highest and most balanced activity, while H-BDP2 and H-BDP3 showed lower levels, with H-BDP2 being the weakest.

To further validate in vivo functionality, the plasmids were microinjected into medaka embryos, and GFP and RFP expression was observed via fluorescence microscopy (Figure 6D). All three fragments successfully drove expression in embryos; however, in some regions, the signals did not fully overlap, suggesting differences in expression levels or spatial distribution. No fluorescence was observed in the negative control group, confirming that the expression was specifically driven by the BDP fragments.

### 2.6. Sequence Analysis of H-BDP Fragments

We then analyzed the sequence characteristics of three histone-related BDP fragments. The lengths of these H-BDPs ranged from 352 to 371 bp, with GC content between 49% and 54%. Core promoter element predictions showed that the TATA box was present in all three fragments, but in H-BDP1 and H-BDP3, it was only near one TSS, with none on the opposite side. In H-BDP3, CCAAT was similarly concentrated on one side. The DPE element appeared frequently across all three fragments (Figure 7, Appendix A). Transcription factor binding site analysis revealed a certain symmetry in their distribution across the three fragments. High-frequency motifs, such as NFY and E2F family motifs, commonly found in human BDPs, were also present in medaka BDPs. Additionally, we found Lhx8 and Tcfl5 binding sites in all three H-BDPs, with the Lhx8 binding site showing similar distribution patterns across the fragments (Figure 8, Appendix A).

## 3. Discussion

The study of bidirectional promoters (BDPs) originated from the analysis of individual gene pairs. Over time, it has become evident that this arrangement is not random but represents a widespread regulatory mechanism. Since Trinklein et al. [1] first identified BDPs in the human genome in 2004, subsequent research has expanded their characterization across diverse species, including archaea, bacteria, yeast, insects, plants, birds, and mammals [4,6,15,16,17]. Divergent gene pairs are now recognized as a common feature across taxa, with their proportions varying significantly. In invertebrates like insects, these pairs range from 1.9% to 25.6% [6], while in mammals such as chimpanzees, dogs, and mice, the proportion is approximately 8% and 9.4% in humans [3]. In this study, we identified that 13% of medaka gene pairs are divergent, aligning with the trend of higher proportions in compact genomes. Neither our study nor Trinklein et al.’s work found a significant correlation between gene density and BDP distribution. This suggests that the arrangement of divergent gene pairs is not simply determined by gene abundance in a genomic region. Instead, evolutionary pressures likely shape this organization to enable the co-regulation of genes involved in essential cellular processes. This clustering facilitates precise transcriptional coordination, which is particularly advantageous in compact genomes, where space constraints necessitate efficient gene organization.

Functional enrichment analysis of medaka BDPs revealed their significant roles in mitochondrial function, RNA processing, and ribosome biogenesis, with strong enrichment in organelle-related processes. These findings align with observations in other species, such as humans, mice, and zebrafish, where BDPs are often associated with housekeeping genes critical for cellular homeostasis [15]. Comparative analyses across species, including humans, chimpanzees, mice, chickens, pufferfish, and zebrafish, have identified numerous conserved BDP regions [15]. These conserved gene pairs are under strong selective pressure to maintain their genomic positions. In mammals, BDPs regulate a wide range of biological processes, particularly catalytic activity and metabolism, whereas in chickens and fish, they are primarily associated with fundamental processes like organelle function and metabolism. These differences reflect species-specific adaptations to distinct evolutionary pressures and ecological contexts, underscoring the critical role of BDPs in coordinating essential biological functions.

Sequence analysis of medaka BDPs revealed that, similar to mammals, they are compact, typically ranging from 0 to 400 bp, with many between 100 and 200 bp. This compact size brings divergent gene pairs closer together, enhancing the influence of shared regulatory elements, such as transcription factor binding sites, on both genes. This spatial proximity allows precise regulation and reduces the number of regulatory elements required, streamlining gene expression and enabling rapid responses to environmental changes. Additionally, medaka BDPs exhibit relatively higher GC content compared to unidirectional promoters (42.06% vs. 39.82%), although both are lower than those in mammals such as humans (66% vs. 53%) [1]. The higher GC content in medaka BDPs likely contributes to promoter stability and efficient transcription factor binding, supporting coordinated gene expression. However, medaka BDP regulation appears to rely less on CpG islands, which are fewer, more dispersed, and less promoter-associated in lower vertebrates like fish and frogs compared to mammals [8]. This suggests a reduced dependency on CpG-mediated regulation in medaka, possibly compensated by alternative regulatory mechanisms.

Analysis of divergent gene pair expression across tissues revealed that co-regulation, including both co-expressed and co-silent gene pairs, is a prominent feature in many tissues. This underscores the role of BDPs in facilitating shared transcriptional control of adjacent genes. However, the variability in co-regulation patterns suggests potential tissue- and sex-specific influences, indicating that the extent of BDP-mediated transcriptional coordination may be shaped by distinct biological contexts. Notably, histone genes in medaka are bidirectionally arranged, with promoters showing strong and consistent transcriptional activity across tissues. As critical components of chromatin, histone genes require synchronized transcription to maintain chromatin stability. The bidirectional arrangement reduces regulatory distance, improving transcription efficiency and enabling rapid responses to cell cycle signals. Our validation experiments confirmed the bidirectional transcriptional activity of three histone-related BDPs, characterized by promoter fragments under 400 bp and GC content above 50%. These features, consistent with typical BDP properties, underline their regulatory efficiency and stability. Similar findings in other species, such as fungi and humans, highlight the synthetic biology potential of histone-derived BDPs. For instance, in Aspergillus, the bidirectional promoter between H4.1 and H3 has been successfully used to express multiple genes related to malformin production, showcasing its utility in multigene expression systems [18].

We further analyzed the core promoter elements and TFBS in histone-related BDPs. TATA-boxes were infrequently observed and often asymmetrically positioned, consistent with findings in humans [8]. While TATA-boxes typically regulate genes with highly variable expression, BDPs are more commonly associated with housekeeping genes requiring stable expression. Instead, motifs like NF-Y and E2F, known to enhance bidirectional transcription and maintain cellular homeostasis [19], were identified in medaka histone BDPs. Additionally, we observed Lhx8 and Tcfl5 motifs, which are critical for germ cell development and spermatogenesis [20,21,22,23,24,25,26], further suggesting potential tissue-specific regulation. These findings highlight the complexity and adaptability of BDP-mediated regulation in medaka, warranting further investigation into their roles across diverse biological contexts.

Nonetheless, this study has certain limitations. The transcriptomic data utilized were restricted to four tissues due to the limited availability of medaka experimental materials, and functional validation was confined to the SG3 cell line. Future investigations should aim to incorporate a broader spectrum of tissues and cell types to further elucidate the regulatory characteristics of BDPs and their potential tissue-specific functionalities. Expanding the diversity of experimental materials will enable a more nuanced understanding of BDP-mediated transcriptional regulation and its biological relevance across different cellular and physiological contexts.

## 4. Materials and Methods

### 4.1. Identification of Divergent Gene Pairs in Medaka

The whole genome sequence of medaka (*Oryzias latipes*) was retrieved from the NCBI Genome database (https://www.ncbi.nlm.nih.gov/datasets/genome/, accessed on 21 May 2022), assembly ASM223467v1). Gene annotation data, including chromosome information, TSS, strand orientation, and gene classifications, were extracted. These annotations were utilized to identify gene pairs that are positioned adjacent to each other on the genome, transcribed in opposite directions, and have TSS separated by less than 1000 base pairs. Such gene pairs were classified as divergent gene pairs (Appendix A).

### 4.2. Divergent Gene Pairs and Gene Density

For each chromosome, the proportion of divergent gene pairs was calculated by dividing the number of divergent gene pairs by the total number of genes on that chromosome. Gene density was then determined by dividing the total number of genes on each chromosome by its length (in megabases, Mb). To assess the relationship between the number of divergent gene pairs and gene density, a Pearson correlation coefficient analysis was performed. Finally, a scatter plot was generated to visually represent the relationship between gene density and the proportion of divergent gene pairs, and a regression analysis was conducted to further evaluate the linear relationship between the two variables.

### 4.3. Functional Analysis of Divergent Gene Pairs

Gene information was extracted from the medaka reference genome using the GenomicFeatures (version 1.56.0) and AnnotationForge (version 1.46.0) packages, and a custom OrgDb database was constructed with the makeOrgPackage function. After loading the database in R (version 4.4.0), Gene Ontology (GO) enrichment analysis of divergent gene pairs was conducted using the clusterProfiler package, covering Biological Process (BP), Cellular Component (CC), and Molecular Function (MF). The Benjamini-Hochberg method was used to adjust *p*-values, with thresholds of 0.01 for *p*-values and 0.05 for *q*-values. The top 10 significant GO terms in each category were visualized using ggplot2. KEGG (Kyoto Encyclopedia of Genes and Genomes) pathway enrichment analysis was also performed using the enrichKEGG function, applying the same thresholds, and significant pathways were visualized.

### 4.4. Expression Analysis of Divergent Gene Pairs

In this study, transcriptome data from wild-type medaka testis and ovary were used alongside publicly available datasets, including liver (GSE252744) and brain (GSE159188), to systematically analyze the expression levels of divergent gene pairs. Public datasets were carefully curated, and only control group samples were included in the analysis to ensure consistency with wild-type conditions, as these samples were not subjected to experimental treatments or genetic modifications.

The testis and ovary datasets were generated in our laboratory using high-quality RNA extracted from three biological replicates per tissue. Strand-specific RNA-seq libraries were constructed following an rRNA-depletion protocol and sequenced on an Illumina NovaSeq 6000 platform (PE150). Clean reads were mapped to the medaka reference genome (ASM223467v1) using Hisat2, with mapping rates exceeding 86%. Detailed quality control metrics for these datasets are provided in the Appendix A.

Genes with an FPKM value greater than 1 were considered expressed, and classification was performed based on the following criteria: when both genes in a divergent gene pair had expression levels below 1, they were defined as “co-silent”; when the expression ratio was within a four-fold range (0.25 to 4), they were defined as “co-expressed”; and when the ratio was below 0.25 or above 4, they were defined as “unidirectionally expressed” [1]. These criteria were consistently applied to analyze each dataset independently, including both laboratory-generated testis and ovary data, as well as publicly available brain and liver datasets. Expression patterns were analyzed within each tissue separately, ensuring the independent evaluation of divergent gene pair distribution in all datasets.

### 4.5. Sequence Characterization of BDP Fragments

BDP fragments were extracted from the regions between the TSS of divergent gene pairs, and their sequences were retrieved from the medaka genome. Additionally, 3000 non-divergent genes were randomly selected, and the 1000 bp upstream of their TSS were extracted as unidirectional promoter fragments. The lengths and GC content of all fragments were then calculated.

### 4.6. Experimental Animals and Sample Collection

Medaka (HdrR strain) used in this study were bred and maintained at our labortotary (College of Jimei University, Xiamen, China) and were reared in a recirculating system at 28 °C with a 14-h light/10-h dark cycle. Fish were fed twice daily, or three times daily during breeding. Before sampling, fish were anesthetized with MS-222 (tricaine methanesulfonate, MedChemExpress, Shanghai Branch, China), and small tail fin clips were collected. Fin clips were treated in 50 mM NaOH at 95 °C for 30 min, neutralized with 1 mM Tris-HCl, and the extracted DNA was stored at −20 °C for later use [27].

### 4.7. Cloning, Transfection, and Microinjection of BDP Fragments

The BDP fragments were amplified by PCR (primers in Table 3) and cloned into the pTol2-GFP-RFP dual-fluorescent reporter plasmid. A plasmid lacking the promoter sequence was used as a negative control. The plasmids were then transfected into SG3 cells, a medaka spermatogonia cell line [28], and red and green fluorescence was observed 48 h later. Additionally, 1-cell stage medaka embryos (15–20 per group) were microinjected with the plasmid (20 ng/μL) [29], and fluorescence was observed 48 h post-injection.

### 4.8. RT-qPCR

Total RNA was extracted from cells using RNAiso Plus (TaKaRa Bio Inc., Kusatsu, Shiga, Japan), and cDNA was synthesized using the HiScript^®^ III 1st Strand cDNA Synthesis Kit (+gDNA wiper) (Vazyme, Nanjing, China). The relative expression levels of GFP and RFP were then measured using the ChamQ Universal SYBR qPCR Master Mix (Vazyme, Nanjing, China). Data analysis was performed using the 2^−△△Ct^ method [30], with the ampicillin resistance gene (*Amp*) on the plasmid serving as an internal control. Statistical analysis was conducted using a *t*-test, with *p* < 0.05 considered statistically significant.

### 4.9. Prediction of Core Promoter Elements and Transcription Factor Binding Sites

The recognition of core promoter elements followed established DNA sequence patterns [8]: TATA box (T[AT]A[AT]A[AGT]), INR ([CT]{2}A[CT][AT][CT]{2}), BRE ([GC]{2}[GA]CGCC), DPE ([AG][CG][AT][CT][ACG][CT]), and CCAAT box ([AG][AG]CCAAT[ACG][AG]). These specific DNA motifs were searched within the target sequences, and their start and end positions, as well as matched sequences, were recorded.

For TFBS prediction, the R packages TFBSTools (version 1.42.0) and JASPAR2018 were used. Motif information for vertebrate transcription factors was obtained from the JASPAR database, and binding site predictions were made based on the position weight matrix (PWM). A match score threshold of 70% or higher was applied to ensure prediction accuracy.

## 5. Conclusions

This study systematically identified BDPs in the medaka genome using bioinformatics analysis, followed by sequence analysis and functional validation of histone-related BDP fragments. Our findings provide critical insights into the regulatory roles of BDPs in fish genomes. Furthermore, this research enhances our understanding of fish genome architecture and offers a theoretical foundation for developing multigene expression systems, which could serve as molecular tools for advancing fish genetic engineering.

## Figures and Tables

**Figure 1 ijms-25-13726-f001:**
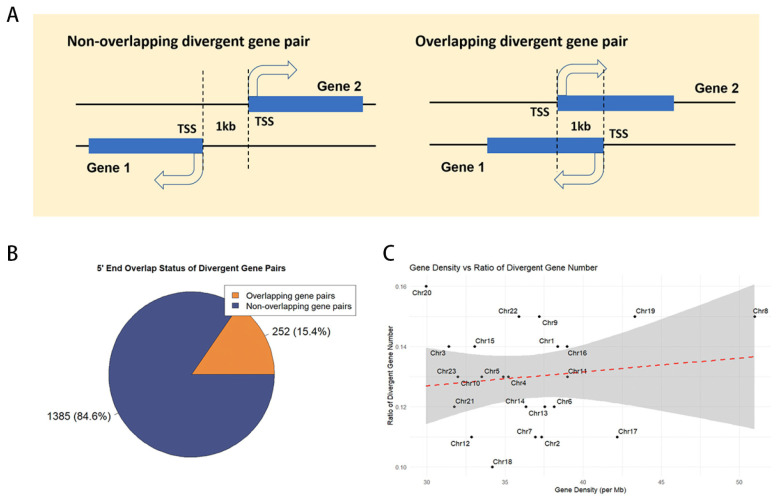
The distribution of divergent gene pairs in the medaka genome. (**A**) Types of divergent gene pairs. TSS, transcription start site. (**B**) Proportion of different types of divergent gene pairs. (**C**) Correlation analysis between the number of divergent gene pairs and gene density.

**Figure 2 ijms-25-13726-f002:**
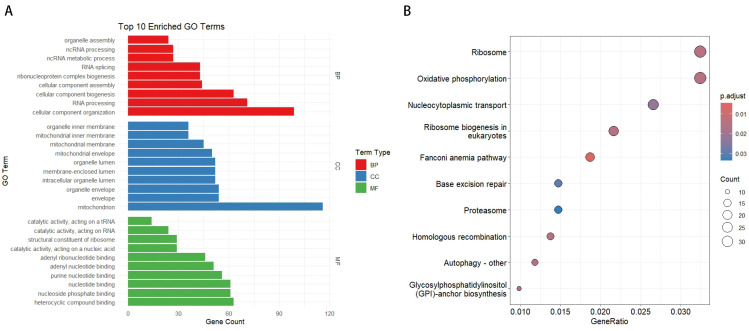
Functional enrichment analysis of divergent gene pairs. (**A**) Top 10 enriched GO terms for Biological Process (BP), Cellular Component (CC), and Molecular Function (MF). The length of each bar represents the number of genes associated with each GO term. (**B**) KEGG pathway enrichment analysis. The y-axis shows the top 10 significantly enriched pathways, while the x-axis represents the gene ratio. The size of the dots indicates the number of genes, and the color gradient shows the adjusted *p*-value.

**Figure 3 ijms-25-13726-f003:**
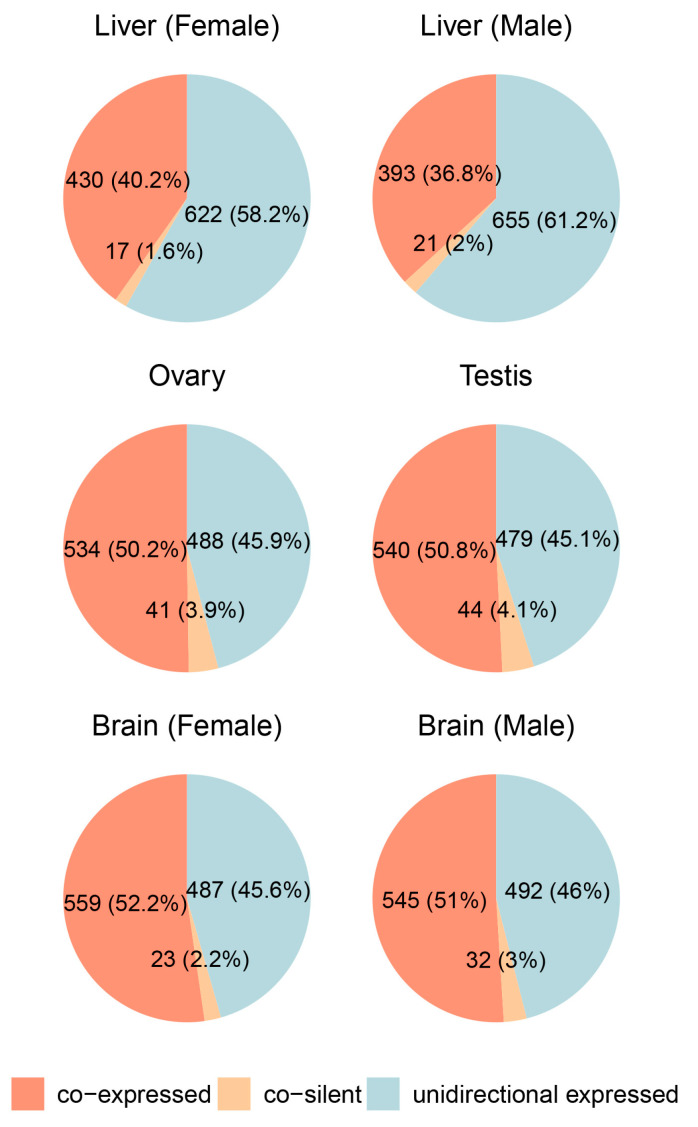
Expression patterns of divergent gene pairs across different tissues in male and female medaka (*Oryzias latipes*).

**Figure 4 ijms-25-13726-f004:**
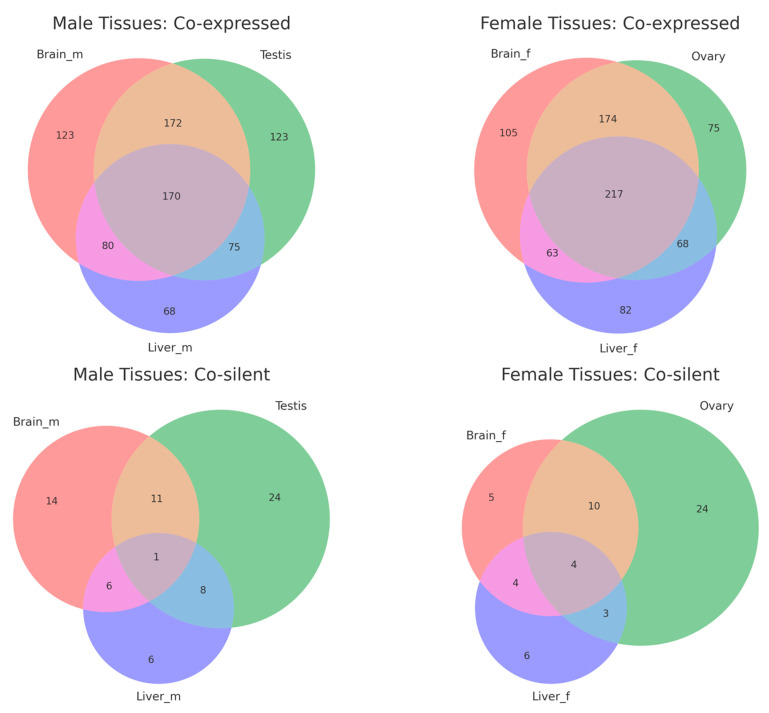
Venn diagrams showing the overlap of co-expressed and co-silent divergent gene pairs across tissues in male and female medaka (*Oryzias latipes*).

**Figure 5 ijms-25-13726-f005:**
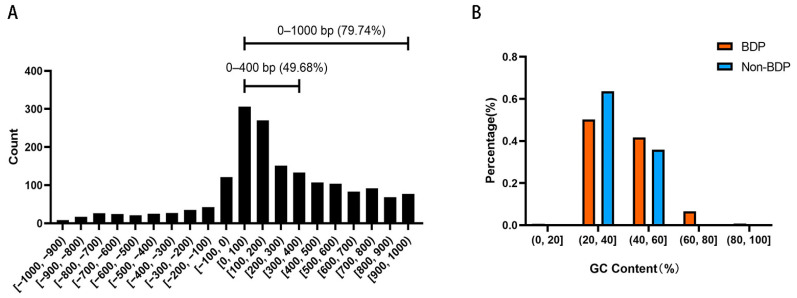
Length and GC content distribution of bidirectional promoter (BDP) fragments. (**A**) BDP fragment length distribution. (**B**) GC content distribution of BDP and non-BDP fragments.

**Figure 6 ijms-25-13726-f006:**
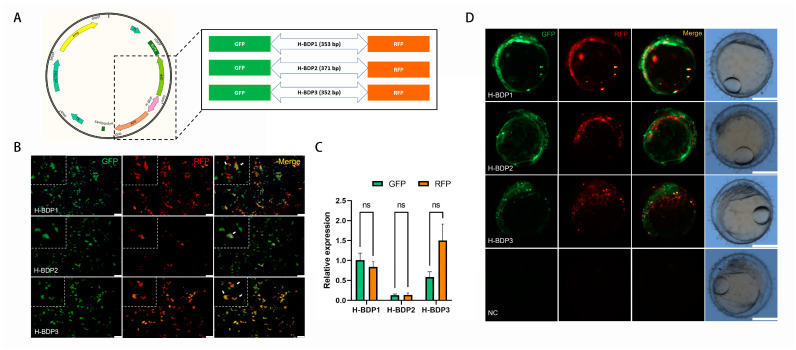
Functional validation of histone bidirectional promoters (H-BDPs). (**A**) Schematic representation of the dual-reporter plasmid construct with GFP and RFP flanking the H-BDP fragments (H-BDP1, H-BDP2, H-BDP3). (**B**) Fluorescence microscopy images showing GFP and RFP expression driven by H-BDP1, H-BDP2, and H-BDP3 in SG3 cells. Green fluorescence represents GFP, red represents RFP, and the merged image shows co-expression. Scale bar, 100 μm. The white arrows indicate cells that simultaneously express RFP and GFP. (**C**) Quantitative analysis of relative GFP and RFP expression levels for H-BDP1, H-BDP2, and H-BDP3 in SG3 cells. ns, not significant. (**D**) Embryonic expression of GFP and RFP under the control of H-BDPs in medaka embryos. Scale bar, 500 μm. NC, negative control, represents embryo injected with promoter-free plasmid.

**Figure 7 ijms-25-13726-f007:**
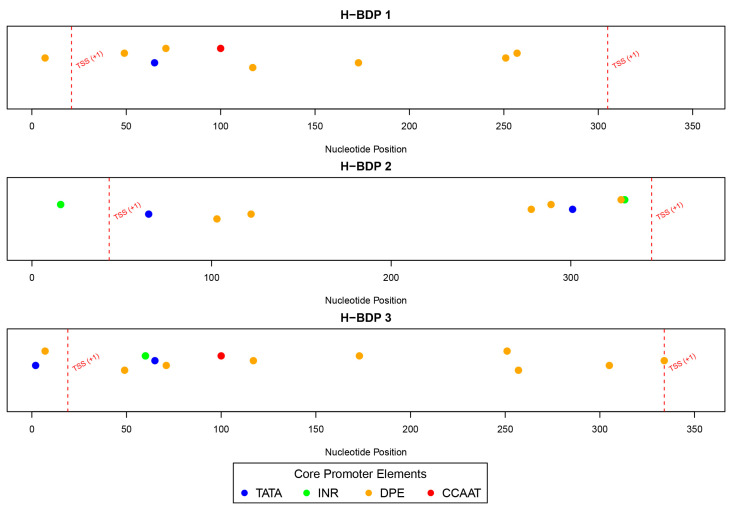
The distribution of core promoter elements (TATA, INR, DPE, and CCAAT) across H-BDP1, H-BDP2, and H-BDP3. Core promoter elements are color-coded and mapped to nucleotide positions. TSS, transcriptional start site.

**Figure 8 ijms-25-13726-f008:**
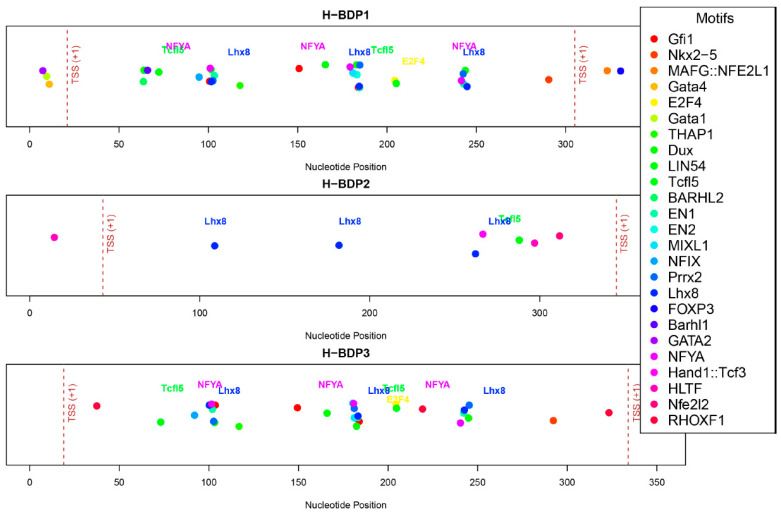
The distribution of predicted transcription factor binding motifs across H-BDP1, H-BDP2, and H-BDP3. Different motifs are shown with distinct colors and mapped to nucleotide positions. TSS, transcriptional start site.

**Table 1 ijms-25-13726-t001:** Distribution of Divergent Genes in the Genome of Medaka (*Oryzias latipes*).

Chromosome	Size (Mb)	Total Gene Number	Gene Density (per Mb)	Divergent Gene Number		
				Pair Number	Gene Number	Ratio
Chr1	37.71	1447	38.37	102	202	0.14
Chr2	25.38	948	37.35	53	105	0.11
Chr3	38.25	1201	31.40	84	167	0.14
Chr4	32.87	1157	35.20	74	147	0.13
Chr5	33.21	1158	34.87	70	140	0.12
Chr6	32.25	1230	38.14	73	146	0.12
Chr7	34.57	1277	36.94	73	146	0.11
Chr8	26.24	1338	50.99	98	195	0.15
Chr9	33.4	1242	37.19	91	182	0.15
Chr10	31.22	1046	33.50	68	136	0.13
Chr11	28.21	1100	38.99	69	138	0.13
Chr12	30.54	1003	32.84	57	114	0.11
Chr13	33.83	1270	37.54	74	148	0.12
Chr14	30.6	1112	36.34	66	132	0.12
Chr15	30.48	1007	33.04	72	179	0.18
Chr16	32.96	1284	38.96	90	179	0.14
Chr17	31.79	1341	42.18	72	143	0.11
Chr18	30.92	1057	34.18	51	102	0.10
Chr19	25.47	1103	43.31	85	169	0.15
Chr20	25.94	777	29.95	61	122	0.16
Chr21	31.15	989	31.75	58	116	0.12
Chr22	28.98	1040	35.89	77	154	0.15
Chr23	24.4	780	31.97	51	101	0.13
Chr24	23.68	926	39.10	67	134	0.14
MT	0.02	37	1850.00	1	2	0.05
Total		26,870		1737	3499	

**Table 2 ijms-25-13726-t002:** Sequence Information and Expression Levels of Histone-Related BDPs.

BDP	Divergent Gene ID	Protein	Sequence			
			Length Between TSSs	GC%	Length Between ATGs	GC%
H-BDP1	ENSORLG00000024975	histone H2A-like	283	57%	353	54%
	ENSORLG00000030596	histone H3				
H-BDP2	ENSORLG00000000884	histone H3	301	58%	371	54%
	ENSORLG00000027232	histone H2A-like				
H-BDP3	ENSORLG00000022956	histone H2A-like	314	52%	352	49%
	ENSORLG00000025490	histone H3				

**Table 3 ijms-25-13726-t003:** Primers.

Primer	Sequence (5′→3′)	Purpose
H-BDP1-F	GTTTAAGGTTCTTATCTGATTCTACAGAGAAG	BDP amplification
H-BDP1-R	TTTACTGACTCGTCTGTCTTTACAGAATC	
H-BDP2-F	CTTCGATGCTTTCTTTCACTTTGCTG	
H-BDP2-R	GTTGTAGTATTCTTTTCTAACTCTGAACAGGAG	
H-BDP3-F	TTTAAAAGATCTTTTCTGATTCTAAAGAGAAGAG	
H-BDP3-R	TTTACTGATTCGTATGTCTTACAGGATC	
GFP-F	TGAGCAAGGGCGAGGAGCTG	RT-qPCR
GFP-R	CCGGACACGCTGAACTTGTGG	
RFP-F	GTGCGCTCCTCCAAGAACGTC	
RFP-R	CTTGGTCACCTTCAGCTTCACGG	
Amp-F	AACTACGATACGGGAGGGCT	
Amp-R	TAGACTGGATGGAGGCGGAT	

## Data Availability

Data is contained within the article and Appendix A.

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
