# Peer review of "Genomic Analysis and Functional Validation of Bidirectional Promoters in Medaka (Oryzias latipes)"

_ijms, 2024, doi:10.3390/ijms252413726_

Round 1
Reviewer 1 Report
Comments and Suggestions for Authors
This study systematically identifies BDPs in the medaka genome using bioinformatics analysis, performed sequence analysis and functional validation of histone-related BDP fragments. This research deepens our understanding of fish genome architecture and provides a theoretical basis for the development of multigene expression systems, which could be published in this journal. However, some details still should be noted.
Question 1: Line 70: "ASM223467v1." please change it into "ASM223467v1".
Question 2: Line 115: What is "M222"? Is it "MS-222"?
Question 3: Line 117: "use [17]" please change it into "use[17]".
Question 4: In Table 2, why is "Chr1" underlined? Is there something specific being emphasized?
Question 5: Line 178: Please check the capitalization of "biological process", "cellular component" and "molecular function" and ensure consistency throughout the manuscript.
Question 6: Line 178: ", and" please change it into "and".
Question 7: In Table 3, why is "ENSORLG00000024975" underlined? Is there something specific being emphasized?
Question 8: Line 272: Remove duplicate word "including".
Question 9: Line 272: The sentence "BDPs exhibit evolutionary conservation across species". This study lacks a comparison of BDP in medaka and other species.
Question 10: Line 279: The sentence "suggesting their arrangement may be more related to gene function than gene count". Please explain the reason.
Question 11: Line 326: The sentence "though these expression patterns may vary depending on tissue specificity". Please specify why only use testis and ovary transcriptome data and only these two tissues this conclusion is a bit arbitrary.
Author Response
Please see the attachment.
We sincerely thank the reviewer for the positive feedback on our study and for highlighting its contributions to understanding fish genome architecture and its potential applications in multigene expression systems. We have carefully addressed the issues raised and made significant revisions to the manuscript to ensure clarity, accuracy, and scientific rigor. Detailed responses to each comment are provided below.
Comment 1: Line 70: "ASM223467v1." please change it into "ASM223467v1".
Response1:Thank you for your comment. We have revised the sentence to include the genome assembly version in parentheses, ensuring clarity and avoiding potential confusion. The updated sentence reads:
“The whole genome sequence of medaka (Oryzias latipes) was retrieved from the NCBI Genome database (https://www.ncbi.nlm.nih.gov/datasets/genome/, assembly ASM223467v1).”
Comment 2: Line 115: What is "M222"? Is it "MS-222"?
Response 2: Thank you for pointing this out. It was indeed a typographical error. "M222" has been corrected to "MS-222" in the revised manuscript.
Comment 3: Line 117: "use [17]" please change it into "use[17]".
Response 3: Thank you for pointing this out. We have removed the unnecessary space and corrected "use [17]" to "use[17]" in the revised manuscript.
Comment 4: In Table 2, why is "Chr1" underlined? Is there something specific being emphasized?
Response 4: Thank you for pointing this out. The underlining of "Chr1" in Table 2 was unintentional and has been removed in the revised manuscript.
Comment 5: Line 178: Please check the capitalization of "biological process", "cellular component" and "molecular function" and ensure consistency throughout the manuscript.
Response 5: Thank you for your comment. We have reviewed the manuscript and ensured consistent capitalization of "Biological Process," "Cellular Component," and "Molecular Function." These terms are now uniformly capitalized throughout the text and figure captions for clarity and consistency.
Comment 6: Line 178: ", and" please change it into "and".
Response 6: Thank you for pointing this out. We have removed the comma and revised ", and" to "and" in the figure legend to improve readability.
Comment 7: In Table 3, why is "ENSORLG00000024975" underlined? Is there something specific being emphasized?
Response 7: Thank you for your comment. The underlining of "ENSORLG00000024975" in Table 3 was unintentional and has been removed in the revised manuscript.
Comment 8: Line 272: Remove duplicate word "including".
Response 8: Thank you for pointing this out. The duplicate word "including" has been removed in the revised manuscript.
Comment 9: Line 272: The sentence "BDPs exhibit evolutionary conservation across species". This study lacks a comparison of BDP in medaka and other species.
Response 9 :Thank you for pointing this out. In the revised manuscript, we have elaborated on the evolutionary conservation of BDPs by integrating comparisons with findings from other species, as discussed in previous studies (e.g., humans, mice, chickens). Specifically, we highlighted that conserved BDP regions across species often regulate genes involved in essential cellular functions, such as metabolism and organelle function, as noted in mammals and teleosts. Furthermore, we emphasized the distinctions between vertebrates and invertebrates, reflecting evolutionary adaptations to distinct ecological niches.
However, we acknowledge that our current study primarily focuses on medaka and does not include direct experimental comparisons of BDPs between medaka and other species. This limitation has been addressed in the Discussion section, where we suggest that future studies could include cross-species analyses of BDPs to better understand their evolutionary and functional divergence.
Comment 10: Line 279: The sentence "suggesting their arrangement may be more related to gene function than gene count". Please explain the reason.
Response 10: Thank you for your comment. We have revised the manuscript to provide a clearer explanation of why BDP arrangement is more related to gene function than gene count. Both our study and previous research in humans found no significant correlation between BDP distribution and gene density. Instead, functional associations between divergent gene pairs, such as co-regulation of housekeeping genes, suggest that evolutionary pressures play a critical role in shaping their arrangement. This revision clarifies the role of functional demands and evolutionary constraints in determining BDP distribution.
Comment 11: Line 326: The sentence "though these expression patterns may vary depending on tissue specificity". Please specify why only use testis and ovary transcriptome data and only these two tissues this conclusion is a bit arbitrary.
Response 11: Thank you for your thoughtful comment. The initial choice of testis and ovary transcriptome data was influenced by the research focus of our group, which primarily investigates reproductive biology. As a result, the transcriptome data we initially analyzed were derived from gonadal tissues, as these were the primary datasets available within our laboratory. Additionally, we chose to rely on our own gonadal data due to our familiarity with the data acquisition and quality control processes, ensuring a high level of confidence in the analyses.
In response to your comment, we have supplemented our analysis with publicly available transcriptome datasets from non-gonadal tissues, including brain (GSE159188) and liver (GSE252744). These additional datasets allowed us to evaluate the expression patterns of divergent gene pairs across multiple tissues, thereby reducing any bias that might arise from focusing solely on gonadal tissues.
To further clarify, we have included a detailed description of the data acquisition process for both our laboratory-generated datasets and the public datasets in the revised Materials and Methods section. We hope these revisions address your concern and provide a more comprehensive perspective on the transcriptional regulation of divergent gene pairs.
These additional datasets allowed us to analyze the expression patterns of divergent gene pairs across a broader range of tissues. The analysis was performed independently for each tissue, and the conclusions have been revised to reflect this broader perspective. The Materials and Methods section has also been updated to include details on the processing and analysis of both laboratory-generated and publicly available transcriptome data.

Reviewer 2 Report
Comments and Suggestions for Authors
Dear Authors,
Overall In my opinion the proposed manuscript is of adequate scientific quality, but there are still some issues, that need to be addressed before its publication can be regarded. The biggest, and most serious issue, which in my opinion at this point is limiting its viability for publication, is reliance on unpublished, and not available data from the prior transcriptome studies. This practice is regarded as golden standard in current molecular biology research. Additionally, no information regarding the method of obtaining this transcriptome data is present, which in itself is very valuable information (for example - GC content bias of Oxford).
In my opinion second issue is that some of the promoters are tissue-specific, and in this case, transcriptome data is coming from ovaries, and testes, fact which should be highlighted.
Thirdly, the Introduction section should be throughly expanded to include additional information, on the BDP's, which includes more information on their role, and specific way of action.
As for minor remarks:
Line 103 - missing space
Line 122 - missing space
Line 124 - missing space
Line 133 - missing space
Line 308 - missing space
Line 314 - missing space
Line 318 - missing space
Line 345 - missing space
Line 357 - missing space
Figures 3, and 6 should be split, and their size should be increased
Author Response
Please see the attachment.
We sincerely thank the reviewer for their thorough review and constructive comments, which have greatly helped us improve the manuscript. In response to the issues raised, we have made significant revisions to the manuscript, including:
- Supplementing the analysis with publicly available transcriptome data from additional tissues (brain and liver) to provide a broader perspective on the expression patterns of divergent gene pairs.
- Providing detailed descriptions of the methods used to obtain and process both laboratory-generated and public transcriptome data.
- Revising relevant sections of the manuscript to clarify the scope of the study and address potential limitations.
We believe these revisions address the reviewer’s concerns and further strengthen the scientific rigor and clarity of the manuscript. Detailed responses to each comment are provided below.
Comment 1: Overall In my opinion the proposed manuscript is of adequate scientific quality, but there are still some issues, that need to be addressed before its publication can be regarded. The biggest, and most serious issue, which in my opinion at this point is limiting its viability for publication, is reliance on unpublished, and not available data from the prior transcriptome studies. This practice is regarded as golden standard in current molecular biology research. Additionally, no information regarding the method of obtaining this transcriptome data is present, which in itself is very valuable information (for example - GC content bias of Oxford).
Response 1
Thank you for pointing out the concerns regarding the reliance on unpublished transcriptome data and the lack of detailed information on its generation. In the revised version of the manuscript, we have made significant updates to address these issues:
Inclusion of Public Datasets: To enhance the comprehensiveness of the study, we have supplemented our analysis with publicly available transcriptome data from brain (GSE159188) and liver (GSE252744). These datasets complement our laboratory-generated testis and ovary datasets and provide a broader perspective on the expression patterns of divergent gene pairs across multiple tissues. Only control group samples were included from these public datasets to ensure consistency with wild-type conditions.
Laboratory-Generated Transcriptome Data: The testis and ovary transcriptome data used in this study are part of a larger dataset generated in our laboratory, which includes both wild-type and genetically edited medaka samples. However, only the wild-type data were utilized in this study to focus on natural transcriptional regulation. For transparency, we have submitted the wild-type transcriptome data to the NCBI database, and they are currently under review. Accession numbers for this subset will be provided upon approval. The genetically edited data will be published separately as part of a different study.
Methodology Details: We have added a detailed description of the procedures used to generate and process the transcriptome data in the Materials and Methods section. This includes RNA sample preparation, library construction (strand-specific RNA-seq protocol with rRNA depletion), sequencing platform (Illumina NovaSeq 6000, PE150), and data quality control metrics (e.g., GC content, mapping rate). The same pipeline was applied to both laboratory-generated and public datasets to ensure consistency in data processing and analysis. Additional quality metrics, such as Q20, Q30, and mapping rates (>86%), are provided in the supplementary materials.
We hope these revisions adequately address your concerns and demonstrate the transparency and scientific rigor of our study.
Comment 2: In my opinion second issue is that some of the promoters are tissue-specific, and in this case, transcriptome data is coming from ovaries, and testes, fact which should be highlighted.
Response 2:
Thank you for your insightful comment. While we acknowledge that some promoters may exhibit tissue-specific expression patterns, the primary objective of this study is to investigate the general expression profiles of divergent gene pairs across multiple tissues, rather than to focus exclusively on the specific regulatory mechanisms in individual tissues. To enhance the comprehensiveness of our analysis, we supplemented our laboratory-generated testis and ovary transcriptome data with publicly available datasets for brain (GSE159188) and liver (GSE252744).
By integrating these datasets, we were able to evaluate the expression patterns of divergent gene pairs across a broader range of tissues. Each tissue was analyzed independently, using consistent classification criteria. While certain tissue-specific promoters were observed, the primary aim of this study remains focused on providing a broader perspective on the transcriptional activity and potential regulatory roles of divergent gene pairs.
We also acknowledge that the limited availability of experimental materials in medaka posed certain constraints on the scope of this analysis. This limitation has been addressed in the revised manuscript, where we note that expanding the range of tissues and experimental materials in future studies will allow for a more comprehensive exploration of tissue-specific regulatory mechanisms and promoter characteristics.
Additionally, detailed descriptions of the data sources and analytical approaches have been included in the Materials and Methods section to ensure transparency and clarity.
Comment 3: Thirdly, the Introduction section should be throughly expanded to include additional information, on the BDP's, which includes more information on their role, and specific way of action.
Response 3:
Thank you for your suggestion. We have revised the Introduction section to include additional details on the roles and mechanisms of BDPs as requested.

Round 2
Reviewer 1 Report
Comments and Suggestions for Authors
I suggest accepting this revised MS.
Reviewer 2 Report
Comments and Suggestions for Authors
The authors have addressed the remarks, and the provided replies, are satisfactory. With the data, mostly available, I dont see any reasons not to suggest the publication of the current version of the manuscript.